# FAIR ADVERSARIAL TRAINING: ON THE ADVERSARIAL ATTACK AND DEFENSE OF FAIRNESS

## ABSTRACT

While numerous work has been proposed to address fairness in machine learning, existing methods do not guarantee fair predictions under imperceptible adversarial feature perturbation, and a seemingly fair model can suffer from large group-wise disparities under such perturbation. Moreover, while adversarial training has been shown to be reliable in improving a model's robustness to defend against adversarial feature perturbation that deteriorates accuracy, it has not been properly studied in the context of adversarial perturbation against fairness. To tackle these challenges, in this paper, we study the problem of adversarial attack and adversarial robustness w.r.t. two terms: fairness and accuracy. From the *adversarial attack* perspective, we propose a unified structure for adversarial attacks against fairness which brings together common notions in group fairness, and we theoretically prove the equivalence of adversarial attacks against different fairness notions. Further, we derive the connections between adversarial attacks against fairness and those against accuracy. From the *adversarial robustness* perspective, we theoretically align robustness to adversarial attacks against fairness and accuracy, where robustness w.r.t. one term enhances robustness w.r.t. the other term. Our study suggests a novel way to unify adversarial training w.r.t. fairness and accuracy, and experiments show our proposed method achieves better robustness w.r.t. both terms.

## 1 INTRODUCTION

As machine learning systems have been increasingly applied in high-stake fields, it is imperative that machine learning models do not reflect real-world discrimination. However, machine learning models have shown biased predictions against disadvantaged groups on several real-world tasks (Larson et al., 2016; Dressel & Farid, 2018; Mehrabi et al., 2021a). In order to improve fairness and reduce discrimination of machine learning systems, a variety of work has been proposed to quantify and rectify bias (Hardt et al., 2016; Kleinberg et al., 2016; Mitchell et al., 2018). Despite the emerging interest in fairness, fairness depreciation in the context of adversarial perturbation and the corresponding defense techniques have not been adequately discussed.

Previous work has demonstrated that by applying small magnitude of adversarial perturbations to input features, the performance (accuracy) of machine learning models can be severely deteriorated (Goodfellow et al., 2014; Madry et al., 2017). For simplicity of discussion, with slight abuse of phraseology we define such perturbation as the **accuracy attack**, i.e., the imperceptible perturbation to deteriorate accuracy, and **accuracy adversarial samples** as samples that are adversarially perturbed by the accuracy attack. We leave the mathematical formulations in Sec. 3.1. In response, work on adversarial training has been proposed to improve robustness of machine learning models to defend against accuracy attacks (Chakraborty et al., 2018; Wong et al., 2020; Sriramanan et al., 2021). We define such robustness as **accuracy robustness**, i.e., a model's ability to resist adversarial perturbation by an **accuracy attack** and remain same predictions on clean and accuracy adversarial samples.

Nonetheless, existing techniques of accuracy attacks cannot be directly applied to the context of fairness depreciation under adversarial perturbation. While accuracy attacks aim at exacerbating the classification error, adversarial attacks against fairness try to deteriorate group-wise parity between different sensitive groups, leading to varied perturbations up to each individual. A successful

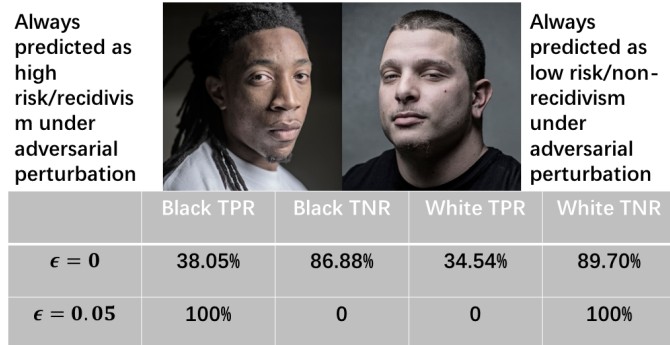

| | Black TPR | Black TNR | White TPR | White TNR |
|---|---|---|---|---|
| $\epsilon = 0$ | 38.05% | 86.88% | 34.54% | 89.70% |
| $\epsilon = 0.05$ | 100% | 0 | 0 | 100% |

Figure 1: Demonstration of adversarial attacks against fairness on COMPAS dataset (Larson et al., 2016) for a statically fair classifier obtained by in-processing (Wang et al., 2022). Under a small perturbation level $\epsilon = 0.05$ ($\leqslant 1.5\%$ of input feature's norm), the disparities in true positive rates (TPR) and true negative rates (TNR) between white and black people increase sharply to $100\%$, and the outcomes of classifier become solely dependent of sensitive information, leading to destructive social injustice against the disadvantaged group (where all the black individuals will be considered as of high risk in this demonstration). This shows that fairness shall not be considered as a static measure, and a classifier with small fairness gaps can show large disparities under fairness attacks. It is important to ensure robustness to fairness attack, but enforcing fairness alone during training does not necessarily improve such robustness. Image credit: (Angwin et al., 2016).

accuracy attack does not necessarily ensure fairness depreciation, and vice versa. Similar to those of accuracy, we define the **fairness attack** as the imperceptible perturbation to deteriorate fairness, and **fairness adversarial samples** as samples that are adversarially perturbed by the fairness attack. We show the mathematical formulations in Sec. 3.2. Work including (Solans et al., 2020; Mehrabi et al., 2021b) first proposed to generate fairness adversarial samples taking into account fairness objectives to perturb the training process and exacerbate bias on clean testing data. However, the detailed mechanism of fairness attacks has not yet been properly discussed, and it remains unclear the relationship between fairness attacks and accuracy attacks.

Just as a model optimized for accuracy in training may not be robust against an accuracy attack, similarly, a fair model trained for static fairness may not inherently possess fairness robustness against fairness attacks. Here we similarly define **fairness robustness** as a model's ability to resist adversarial perturbation by an **fairness attack** and remain same predictions on clean and fairness adversarial samples. As shown in Fig. 1, fairness can be volatile under adversarial perturbations, where a small degree of perturbation can lead to significant variations in group-wise disparities, and enforcing fairness alone during training does not necessarily lead to improvement in robustness against fairness attacks. It is worth noting that fairness attacks can be employed to depreciate the trustworthiness of models and aggravate discrimination against disadvantaged groups, fostering social division and social in-cohesion. Hence it is imperative to improve a model's robustness to fairness attacks. As discussed above, adversarial training has been shown successful in improving accuracy robustness. In fairness literature, although adversarial learning has been widely discussed, most of them have been focused on applying adversarial learning as a means to unlearn the impact of sensitive attributes to achieve static fairness (Madras et al., 2018; Creager et al., 2019). Chhabra et al. (2022) first propose a defense framework for adversarial perturbation against fairness; however, such perturbation is targeted against sensitive information, rather than input features. We therefore consider two questions:

- *Can we formulate adversarial attacks against various fairness notions?*
- *Can we propose adversarial training techniques to defend against such fairness attacks?*

In this work, we propose a general framework for fairness attacks, where we show impacts of fairness attacks up to each individual under different notions, as well as the connections between these notions regarding gradient-based attacks. Based on this unified framework, we discuss the relationship

between fairness attacks and accuracy attacks. Furthermore, we show that despite the discrepancies in adversarial perturbations between the fairness attack and the accuracy attack for certain samples, fairness robustness and accuracy robustness do not necessarily conflict with each other. Based on the spatial proximity between such samples and samples where the fairness attack and the accuracy attack acts in the same direction, we show theoretically how fairness robustness and accuracy robustness can benefit from each other, i.e., the alignment between fairness robustness and accuracy robustness. Our theoretical results suggest a novel defense framework, *fair adversarial training*, which incorporates fair classification with adversarial training so as to improve fairness robustness. We summarize our contribution as follows:

- We propose a unified framework for fairness attacks, which brings together different notions in group fairness.

- We theoretically demonstrate the alignment between fairness robustness and accuracy robustness, and we propose a novel defense framework, *fair adversarial training*, which incorporates fairness robustness with fair classification.

- We empirically validate the superiority of our method under fairness attacks, and the connection between fairness robustness and accuracy robustness on four benchmark datasets.

## 2 RELATED WORK

### 2.1 FAIRNESS IN MACHINE LEARNING

Fairness has gained much attention in machine learning society. Different notions have been proposed to quantify discrimination of machine learning models, including individual fairness (Lahoti et al., 2019; John et al., 2020; Mukherjee et al., 2020), group fairness (Feldman et al., 2015; Hardt et al., 2016; Zafar et al., 2017) and counterfactual fairness (Kusner et al., 2017). Our work is most closely related with group fairness notions. Works on group fairness generally fall into three categories: preprocessing (Creager et al., 2019; Jiang & Nachum, 2020; Jang et al., 2021), where the goal is to adjust training distribution to reduce discrimination; in-processing (Zafar et al., 2017; Jung et al., 2021; Roh et al., 2021), where the goal is to impose fairness constraint during training by reweighing or adding relaxed fairness regularization; and post-processing (Hardt et al., 2016; Jang et al., 2022), where the goal is to adjust the decision threshold in each sensitive group to achieve expected fairness parity.

### 2.2 ADVERSARIAL MACHINE LEARNING

Adversarial training and adversarial attack have been widely studied in trustworthy machine learning. Goodfellow et al. (2014) propose a simple one-step gradient-based attack to adversarially perturb the input features. Madry et al. (2017) extend the one-step attack to an iterative attack strategy and show that iterative strategy is better at finding accuracy adversarial samples. Accordingly, different methods on adversarial defense have been proposed (Shafahi et al., 2019; Wong et al., 2020; Xie et al., 2020; Cui et al., 2021; Jia et al., 2022) to improve accuracy robustness of a classifier. However, few literature has addressed adversarial training and attack against fairness. Some work discusses the problem of fairness poisoning attack during training (Solans et al., 2020; Mehrabi et al., 2021b); however, it is not clear how fairness attack would influence the predicted soft labels, and the relationship between the fairness attack and the accuracy attack, as well as the corresponding adversarial robustness remains unclear.

### 2.3 FAIRNESS IN ADVERSARIAL ROBUSTNESS

Group fairness in the context of adversarial robustness has been less studied in current work. Work including (Nanda et al., 2021; Xu et al., 2021; Ma et al., 2022) argues that adversarial training without proper regularization leads to class-wise disparities in accuracy and robustness. However, group-wise disparities are not considered in these work, and the formulation of disparities by these work is not in accord with notions in group fairness. Recent work studies the poisoning attack against group fairness measure (Solans et al., 2020; Mehrabi et al., 2021b); however, these work lacks in-depth discussion regarding the detailed mechanism of adversarial attacks against fairness, and no

defense technique is considered in these work. Our work is most related to Chhabra et al. (2022), where adversarial perturbation against sensitive information and the corresponding defense mechanism are considered. In comparison, our framework considers feature-level perturbation, rather than sensitive-information-level.

# 3 PROBLEM DEFINITION

## 3.1 ADVERSARIAL ATTACK AGAINST ACCURACY

We start by formulating the **accuracy attack**. Denote $x \in \mathbb{R}^n$ as the input feature, $y \in \{0, 1\}$ as the label, and $a \in \{0, 1\}$ as the sensitive attribute. Let $f : \mathbb{R}^n \to [0, 1]$ be the function of classifier, then the objective of accuracy attack for sample $(x, y, a)$ can be formulated as

$$\arg\max_{\epsilon} L_{\text{CE}}(f(x + \epsilon), y), \text{ s.t.} \|\epsilon\| \leq \epsilon_0,$$

where $\|\epsilon\|$ refers to the $L^p$ norm of $\epsilon$ with a general choice of $L^\infty$ norm, and $L_{\text{CE}}$ is the cross-entropy loss. A common way to obtain accuracy adversarial samples is through projected gradient descent (PGD) attack, where **accuracy adversarial samples** are updated in each step based on the signed gradient:

$$x^{t+1} = \Pi_{x+S} \left( x^t + \alpha \operatorname{sign} \left( \nabla_x L_{\text{CE}}(x, y) \right) \right),$$

where $\alpha$ is the step size, and $S := \{\epsilon, \|\epsilon\| \leq \epsilon_0\}$ is the set of allowed perturbation. PGD attack has been shown to be effective in finding adversarial samples compared with one-step adversarial attack (Madry et al., 2017).

## 3.2 ADVERSARIAL ATTACK AGAINST FAIRNESS

**Fairness attack** has yet been less studied in current literature. Inspired by accuracy attack, we propose to formulate the fairness attack as follows:

$$\arg\max_{\epsilon} L(f(x + \epsilon), a, y), \text{ s.t.} \|\epsilon\| \leq \epsilon_0,$$

where $L$ is some relaxed fairness constraint. We consider two widely adopted group fairness notions: disparate impact (DI) and equalized odds (EOd). For a testing set $\mathbb{S} = \{(x_i, y_i, a_i), 1 \leq i \leq N\}$, denote $\mathbb{S}_{jk} := \{x_i | y_i = j, a_i = k\}$, and $\mathbb{S}_{.k} := \{x_i | a_i = k\}$. The relaxations for fairness attacks corresponding to DI and EOd (Madras et al., 2018; Wang et al., 2022) can be formulated as:

$$L_{\text{DI}} = \left| \sum_{x_i \in \mathbb{S}_{.1}} \frac{f(x_i)}{|\mathbb{S}_{.1}|} - \sum_{x_i \in \mathbb{S}_{.0}} \frac{f(x_i)}{|\mathbb{S}_{.0}|} \right|, L_{\text{EOd}} = \sum_y \left| \sum_{x_i \in \mathbb{S}_{y0}} \frac{f(x_i)}{|\mathbb{S}_{y0}|} - \sum_{x_i \in \mathbb{S}_{y1}} \frac{f(x_i)}{|\mathbb{S}_{y1}|} \right|. \quad (1)$$

And **fairness adversarial samples** can be obtained analogous to the accuracy attack via PGD attack:

$$x^{t+1} = \Pi_{x+S} \left( x^t + \alpha \operatorname{sign} \left( \nabla_x L(x, a, y) \right) \right).$$

# 4 CONNECTION BETWEEN THE FAIRNESS ATTACK AND THE ACCURACY ATTACK

Before going to the discussion, we first clarify the notations. We denote $x_{\text{sub},a}^{t,\text{obj}}$ as the adversarial sample(s) generated from the clean subgroup $\{\text{sub}, a\}$ at $t$-th iteration targeting attack type obj $\in \{\text{DI}, \text{EOd}, \text{Acc}\}$. For example, $x_{\text{TP},0}^{t,\text{Acc}}$ refers to the accuracy adversarial sample(s) generated form the true positive (TP) sample(s) in the disadvantaged group ($a = 0$) at $t$-th iteration. We will slightly abuse the notation $x_{\text{sub},a}^{t,\text{obj}}$ in this section to denote both individual samples and a set of samples within a subgroup. We denote $x_{\text{sub},a}^{\text{obj}}$ as the adversarial sample(s) obtained after the attack type obj and $x_{\text{sub},a}$ as the clean samples. For example, $x_{\text{FP},1}$ means clean false positive samples in the sensitive group $a = 1$. We denote as $p_{\text{sub},a}^{\text{obj}}$ the distribution of soft predictions in the clean subgroup $\{\text{sub}, a\}$ after the attack type obj and $p_{\text{sub},a}$ the distribution of soft predictions in the clean subgroup $\{\text{sub}, a\}$ without attack. Without loss of generality, we assume $a = 1$ is the advantaged group[1]. We refer to fairness attacks targeting DI and EOd as DI attack and EOd attack, respectively.

---

[1]Here we define the advantaged group as the one with higher average positive prediction.

### 4.1 Connection between EOd and DI attack

We now discuss the detailed relationship between DI and EOd attack. The following corollary states the compatibility of the two objectives:

**Corollary 1.** *The adversarial objective of EOd attack is lower-bounded by that of DI attack up to multiplicative constants.*

We defer the proof to appendix. Corollary 1 shows the connection between adversarial attacks against different group fairness notions, where these attacks perturb the predicted soft labels against sensitive attributes. Specifically, a successful DI attack also leads to a successful EOd attack, while the opposite does not necessarily hold true. In light of this, we will focus on **DI attack** as means of the fairness attack for the rest of this paper. In the following context, we refer to DI attack as fairness attack unless otherwise specified.

For a given sample $(x_j, y_j, a_j)$ in the advantaged group, we can rewrite $L_{\mathrm{DI}}$ in equation 1 as:

$$L_{\mathrm{DI}} = \left| \sum_{x_i \in \mathbb{S}_{.1}} \frac{f(x_i)}{|\mathbb{S}_{.1}|} - \sum_{x_i \in \mathbb{S}_{.0}} \frac{f(x_i)}{|\mathbb{S}_{.0}|} \right| = \frac{f(x_j)}{|\mathbb{S}_{.a_j}|} + \sum_{x_i \in \mathbb{S}_{.a_j} \setminus \{x_j\}} \frac{f(x_i)}{|\mathbb{S}_{.a_j}|} - \sum_{x_i \in \mathbb{S}_{.\hat{a}_j}} \frac{f(x_i)}{|\mathbb{S}_{.\hat{a}_j}|} = \frac{f(x_j)}{|\mathbb{S}_{.a_j}|} + C_j,$$

(2)

where $\hat{a}_j = |1 - a_j|$ and $C_j$ is a constant w.r.t. $x_j$ since it does not affect $\frac{\partial L_{\mathrm{DI}}}{\partial x_j}$. This shows that the DI attack is expected to *maximize* the prediction in the advantaged group. Similarly, for a sample $(x_k, y_k, a_k)$ in the disadvantaged group, we have:

$$L_{\mathrm{DI}} = \left| \sum_{x_i \in \mathbb{S}_{.1}} \frac{f(x_i)}{|\mathbb{S}_{.1}|} - \sum_{x_i \in \mathbb{S}_{.0}} \frac{f(x_i)}{|\mathbb{S}_{.0}|} \right| = -\frac{f(x_k)}{|\mathbb{S}_{.a_k}|} + C_k,$$

(3)

where $C_k$ is a constant w.r.t. $x_k$ thus does not affect $\frac{\partial L_{\mathrm{DI}}}{\partial x_k}$. equation 3 shows that the DI attack in the disadvantaged group is contrary to that of advantaged group, where the predictions are expected to be *minimized*.

### 4.2 Connection between the fairness attack and the accuracy attack

We move on to discuss the connection between the fairness attack and the accuracy attack. The following corollary shows the connection between the fairness attack and the accuracy attack:

**Corollary 2.** *The fairness attack and the accuracy attack operate in the same direction regarding true negative (TN) and false positive (FP) samples in the advantaged group and true positive (TP) and false negative (FN) samples in the disadvantaged group.*

We defer the detailed proof to appendix. Notably, the fairness attack and the accuracy attack behave in the opposite direction for the remaining sets of samples (i.e., TP and FN samples in the advantaged group, and TN and FP samples in the disadvantaged group). Specifically, for the two subgroups $x_{\mathrm{TP},1}$ and $x_{\mathrm{TN},0}$, the fairness attack aims at maximizing their predicted soft labels as in equation 2 and equation 3, respectively. This results in maximizing the predicted soft labels for $x_{\mathrm{TP},1}$ and minimizing the predicted soft labels for $x_{\mathrm{TN},0}$. Whereas the accuracy attack seeks to minimize the predicted soft labels for $x_{\mathrm{TP},1}$ and maximize the predicted soft labels for $x_{\mathrm{TN},0}$.

Likewise, for the subgroups $x_{\mathrm{FN},1}$ and $x_{\mathrm{FP},0}$, the fairness attack tries to 'correct' the predicted soft labels such that the adversarial predictions align with the ground-truth labels. In contrast, the accuracy attack is designed to exacerbate the error within these subgroups. We summarize the connection between the the fairness attack and the accuracy attack on various subgroups in Table 1.

| Sensitive Group | Same Direction | Opposite Direction |
|---|---|---|
| Disadvantaged ($a = 0$) | $x_{\mathrm{TP},0}, x_{\mathrm{FN},0}$ | $x_{\mathrm{TN},0}, x_{\mathrm{FP},0}$ |
| Advantaged ($a = 1$) | $x_{\mathrm{TN},1}, x_{\mathrm{FP},1}$ | $x_{\mathrm{TP},1}, x_{\mathrm{FN},1}$ |

Table 1: Connection between the fairness attack and the accuracy attack regarding samples in different subgroups.

# 5 ALIGNMENT BETWEEN FAIRNESS ROBUSTNESS AND ACCURACY ROBUSTNESS

We now discuss the alignment between fairness robustness and accuracy robustness. According to Table 1, the relationship between fairness robustness and accuracy robustness is straightforward on the four subgroups in 'Same Direction' category. Since the the fairness attack and the accuracy attack operate in the same direction for those samples, the fairness robustness and accuracy robustness also attain alignment on these samples. Consider sample $x_i$ from the 'Same Direction' groups, by Corollary 2 we have:

$$f(x_i^{t,\text{Fair}}) = f\left(\Pi_{x_i+S}\left(x_i^{t-1} + \alpha\,\text{sign}\left(\nabla_{x_i}L_{\text{DI}}(x_i, a_i, y_i)\right)\right)\right)$$
$$= f\left(\Pi_{x_i+S}\left(x_i^{t-1} + \alpha\,\text{sign}\left(\nabla_{x_i}L_{\text{CE}}(x_i, y_i)\right)\right)\right).$$

Under same perturbation level $\epsilon$ and same step size $\alpha$ up to $T$ iterations, the fairness attack and the accuracy attack leads to equivalent perturbations in soft predictions regarding $x_i$'s in the 'Same Direction' category. Therefore, it is feasible to leverage existing adversarial training tools targeting accuracy robustness to improve fairness robustness for such samples. However, such alignment cannot be directly extended to the four subgroups in the 'Opposite Direction' category. As the fairness attack and the accuracy attack operate in the opposite direction, it is not straightforward whether there exists alignment or misalignment between fairness robustness and accuracy robustness regarding those samples.

Therefore, in the following we focus our discussion on the four 'Opposite Direction' subgroups in Table 1: $x_{\text{TP},1}$, $x_{\text{FN},1}$, $x_{\text{TN},0}$, $x_{\text{FP},0}$. Under $\epsilon$-level fairness attack with step size $\alpha$ and up to $T$ iterations, we define $D_{\text{sub},a}^{\text{Fair}} := |L_{\text{CE}}(f(x_{\text{sub},a}^{\text{Fair}}), y) - L_{\text{CE}}(f(x_{\text{sub},a}), y)|$ as the change of cross-entropy loss for sample $x_{\text{sub},a}$ and $\delta_{\text{sub},a}^{\text{Fair}} := |f(x_{\text{sub},a}^{\text{DI}}) - f(x_{\text{sub},a})|$ as the change of $f(x_{\text{sub},a})$. Therefore, $D_{\text{sub},a}^{\text{Fair}}$ and $\delta_{\text{sub},a}^{\text{Fair}}$ are related with **fairness robustness**, and smaller $D_{\text{sub},a}^{\text{Fair}}$ and $\delta_{\text{sub},a}^{\text{Fair}}$ indicate better fairness robustness. Likewise, under $\epsilon$-level accuracy attack with step size $\alpha$ and up to $T$ iterations, we define $D_{\text{sub},a}^{\text{Acc}} := |L_{\text{CE}}(f(x_{\text{sub},a}^{\text{Acc}}), y) - L_{\text{CE}}(f(x_{\text{sub},a}), y)|$ as the change of cross-entropy loss for sample $x_{\text{sub},a}$ and $\delta_{\text{sub},a}^{\text{Acc}} := |f(x_{\text{sub},a}^{\text{Acc}}) - f(x_{\text{sub},a})|$ as the change of $f(x_{\text{sub},a})$, and smaller $D_{\text{sub},a}^{\text{Acc}}$ and $\delta_{\text{sub},a}^{\text{Acc}}$ indicates better **accuracy robustness**. Before we state the detailed relationship, we first state the assumptions we need to prove the relationship:

**Assumption 1.** *The gradient of $f$ w.r.t. input feature $x$ is Lipschitz with constant $K$.*

**Assumption 2.** *The distributions $p_{sub,a}$ are uniformly bounded by constants $M_{sub,a}$.*

Under Assumption 1 and 2, below we discuss the alignment between fairness robustness and accuracy robustness in two directions, i.e., how fairness/accuracy robustness improves accuracy/fairness robustness. While the assumption of Lipschitz gradient seems a bit strong, it is a widely used assumption for neural network, and it is feasible to estimate the Lipschitz constant (Fazlyab et al., 2019; Shi et al., 2022). Also, the difference in fairness/accuracy robustness as discussed in the Theorem 1 and 3 are indeed upper-bounded by the Lipschitz constant $K$, and a smaller $K$ indicates better upper-bounds for the difference in robustness, which also suggests better alignment between fairness robustness and accuracy robustness.

## 5.1 FROM ACCURACY ROBUSTNESS TO FAIRNESS ROBUSTNESS

We first derive the guarantee for fairness robustness by accuracy robustness. We will focus on $x_{\text{FN},1}$ and $x_{\text{FP},0}$, as fairness attack regarding $x_{\text{TP},1}$ and $x_{\text{TN},0}$ does not affect fairness, i.e., the predicted labels for $x_{\text{TP},1}$ and $x_{\text{TN},0}$ will remain the same before and after the fairness attack.

**Theorem 1.** *Given a classifier $f$, consider $\epsilon$-level fairness attack with step size $\alpha$ and up to $T$ iterations, the difference of fairness robustness between $x_{FN,1}$ and $x_{FN,0}$ is upper-bounded by the accuracy robustness of $x_{FN,0}$ up to an additive and a multiplicative constant:*

$$D_{FN,1}^{Fair} \le \min_{x_{FN,0}\in\mathbb{S}_{10}} D_{FN,0}^{Acc} + \alpha\sum_{t=1}^{T}G_t,$$

$$G_t = \left[ \frac{\sqrt{n} K d(x_{FN,1}^{t-1,Fair}, x_{FN,0}^{t-1,Fair})}{f(x_{FN,1}^{t-1,Fair})} + \eta_t \delta_{FN,0}^{t-1,Acc} \right], \eta_t = \left| \frac{f(x_{FN,0}^{t-1,Fair}) - f(x_{FN,1}^{t-1,Fair})}{f(x_{FN,1}^{t-1,Fair}) f(x_{FN,0}^{t-1,Fair})} \right|.$$

Detailed proof and empirical verification can be found in the appendix. As discussed in Section 4.2, adversarial training w.r.t. accuracy also improves fairness robustness of subgroup $x_{FN,0}$ while it is unclear for subgroup $x_{FN,1}$. Thus, we leverage $x_{FN,0}$ to explore robustness guarantee against fairness attack for $x_{FN,1}$.

Specifically, for $f'$ under adversarial training w.r.t. accuracy and $f$ under normal training, we have similar upper-bound, except that we now have $\delta_{FN,0}^{'t-1,Acc} \leq \delta_{FN,0}^{t-1,Acc}$, which indicates a tighter upper-bound for $f'$ in Theorem 1. For the marginal advantaged FN samples ($x_{FN,1}$) which are more vulnerable under the fairness attack, we have their fairness robustness bounded by marginal disadvantaged FN samples ($x_{FN,0}$), and smaller $\delta_{FN,0}^{Acc}$, or tighter bound indicates better fairness robustness. Similar inequality in Theorem 1 also holds for $x_{FP,0}$ and $x_{FP,1}$:

*Remark* 1. For $x_{FP,0}$ and $x_{FP,1}$, we have similar inequality regarding the upper-bound of robustness difference:

$$D_{FP,0}^{Fair} \leq \min_{x_{FP,1} \in \mathbb{S}_{01}} D_{FP,1}^{Acc} + \alpha \sum_{t=1}^{T} H_t,$$

$$H_t = \left[ \frac{\sqrt{n} K d(x_{FP,0}^{t-1,Fair}, x_{FP,1}^{t-1,Fair})}{f(x_{FP,0}^{t-1,Fair})} + \rho_t \delta_{FP,1}^{t-1,Acc} \right], \rho_t = \left| \frac{f(x_{FP,0}^{t-1,Fair}) - f(x_{FP,1}^{t-1,Fair})}{f(x_{FP,1}^{t-1,Fair}) f(x_{FP,0}^{t-1,Fair})} \right|.$$

## 5.2 FAIR ADVERSARIAL TRAINING

Theorem 1 provides robustness guarantee in terms of changes in soft predictions under the fairness attack regarding 'Opposite Direction' samples. Based on such discussion, we now derive the fairness robustness guarantee regarding fairness measures, namely DI and EOd. Consider $DI^{Fair}$, DI, $EOd^{Fair}$ and EOd as the fairness measures after and before the fairness attack, the following theorem states the fairness guarantee by static fairness and accuracy robustness under the fairness attack:

**Theorem 2.** *Given a classifier $f$, consider $\epsilon$-level fairness attack with step size $\alpha$ and up to $T$ iterations, let $\Delta_{sub,a}^{obj} := \max_{\{x_{sub,a} \in \mathbb{S}_{sub,a}\}} \delta_{sub,a}^{obj}$ be the maximum shift in soft predictions within the subgroup under attack type obj, the resulting fairness measures are upper-bounded by the corresponding clean measures and accuracy robustness of the classifier up to a multiplicative constant:*

$$DI^{Fair} \leq DI + M(\Delta_{TP,0}^{Acc} + \min_{j \in \mathbb{S}_{FP,1}} (D_j^{Acc} + H_j) + \Delta_{TN,1}^{Acc} + \min_{j \in \mathbb{S}_{FN,0}} (D_j^{Acc} + G_j)),$$

$$EOd^{Fair} \leq EOd + M((\Delta_{TP,0}^{Acc} + \min_{j \in \mathbb{S}_{FP,1}} (D_j^{Acc} + H_j) + \Delta_{TN,1}^{Acc} + \min_{j \in \mathbb{S}_{FN,0}} (D_j^{Acc} + G_j))),$$

$$M = \max_{\{sub,a\}} M_{sub,a}.$$

We defer full proof to the Appendix. The DI and EOd terms in the upper-bounds of Theorem 2 corresponds to the static fairness, and the remainder corresponds to the accuracy robustness as stated in Theorem 1. Consequently, enhancing fairness during training results in lower values of DI and EOd, and enhancing accuracy robustness results in lower value of the remainder, leading to smaller upper-bounds for $DI^{Fair}$ and $EOd^{Fair}$, i.e., smaller fairness violation under the fairness attack.

One direct result regarding Theorem 2 is to incorporate accuracy adversarial samples during training to obtain a classifier that is also robust to fairness attack. Thus, we consider the following to minimize the fairness gap while ensuring accuracy robustness, as means to ensure fairness robustness. Specifically, we propose to improve fairness robustness of classifier by incorporating accuracy adversarial samples and fairness constraints during training:

$$\arg\min_{\theta} \frac{1}{N} \sum_{i=1}^{N} L_{CE}(f(x_i^{Acc}), y_i), \ s.t. \ L \leq \gamma, \tag{4}$$

where $x_i^{Acc}$ is the accuracy adversarial sample by $x_i$, the $L_{CE}(f(x_i^{Acc}), y_i)$ term corresponds to accuracy robustness, and $L \leq \gamma$ corresponds to static fairness, as we derived in Theorem 2. $L$ can be explicitly specified by fairness relaxations during training or implicitly specified as preprocessing or post-processing techniques.

### 5.3 FROM FAIRNESS ROBUSTNESS TO ACCURACY ROBUSTNESS

For the other direction, under Assumption 1, we have the following guarantee for accuracy robustness by fairness robustness. We will focus on $x_{\text{TP},1}$ and $x_{\text{TN},0}$, as accuracy attack regarding $x_{\text{FN},1}$ and $x_{\text{FP},0}$ does not affect accuracy, i.e., the predicted labels remain false before and after the accuracy attack.

**Theorem 3.** *Given a classifier $f$, consider $\epsilon$-level accuracy attack with step size $\alpha$ and up to $T$ iterations, the accuracy robustness of $x_{TP,1}$ is upper-bounded by the fairness robustness of $x_{TP,0}$ up to an additive constant:*

$$\delta_{TP,1}^{Acc} \leq \min_{x_{TP,0} \in \mathbb{S}_{10}} \delta_{TP,0}^{Fair} + \sum_{t=1}^{T} \sqrt{n}\alpha K d(x_{TP,1}^{t-1,Acc}, x_{TP,0}^{t-1,Acc}).$$

Here the the fairness attack and the accuracy attacks are in alignment regarding $x_{\text{TP},0}$, which we use to upper-bound accuracy robustness of $x_{\text{TP},1}$. Specifically, the first term in the RHS of the inequality corresponds to the fairness robustness of $x_{\text{TP},0}$, and the second term is determined by the spatial distance between $x_{\text{TP},0}$ and $x_{\text{TP},1}$ under the accuracy attack. Theorem 3 shows that adversarial training w.r.t. fairness also benefits accuracy robustness. Specifically, for $f''$ under adversarial training w.r.t. fairness and $f$ under normal training, we have similar inequality, except that we now have $\delta_{\text{TP},0}^{''\text{Acc}} \leq \delta_{\text{TP},0}^{\text{Acc}}$, which indicates better accuracy robustness for $x_{\text{TP},1}$ under adversarial training. Similar upper-bound also holds for TN samples:

*Remark* 2. For $x_{\text{TN},1}$ and $x_{\text{TN},0}$, we have similar inequality regarding the upper-bound of accuracy robustness:

$$\delta_{\text{TN},0}^{\text{Acc}} \leq \min_{x_{\text{TN},1} \in \mathbb{S}_{01}} \delta_{\text{TN},1}^{\text{Fair}} + \sum_{t=1}^{T} \sqrt{n}\alpha K d(x_{\text{TN},0}^{t-1,\text{Acc}}, x_{\text{TN},1}^{t-1,\text{Acc}}).$$

Since the change of predictions under accuracy attack is upper-bounded by fairness robustness, it is also feasible to improve accuracy robustness of classifier by using fairness adversarial samples during training.

## 6 EXPERIMENTS

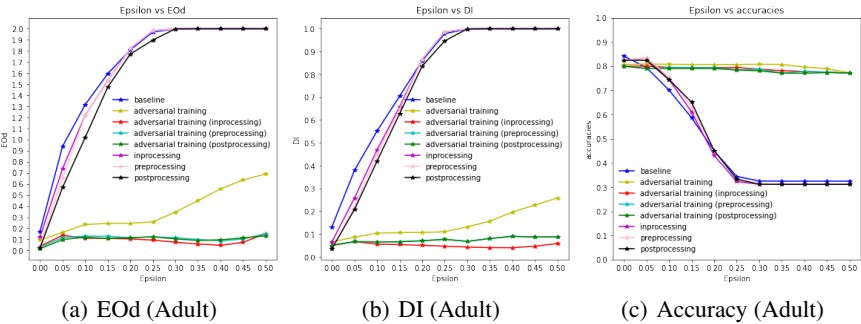

|     |     |     |
| :-: | :-: | :-: |
| (a) EOd (Adult) | (b) DI (Adult) | (c) Accuracy (Adult) |

Figure 2: Change of accuracy, DI and EOd under fairness attack on Adult dataset. Our adversarial training methods (preprocessing, in-processing, post-processing) obtain improved fairness robustness (lower EOd, DI and higher accuracy) with significant margin.

We evaluate our method on four datasets: Adult (Dua & Graff, 2017), COMPAS (Larson et al., 2016), German (Dua & Graff, 2017) and CelebA (Liu et al., 2015). Details of datasets and experimental setup are in the Appendix. We use accuracy as performance evaluation, and disparate impact (DI) and equalized odds (EOd) as fairness metric. In the following subsections, we validate the adversarial training framework under the fairness attack and the accuracy attack, respectively.

### 6.1 ROBUSTNESS AGAINST FAIRNESS ATTACK

We consider five different methods for comparison: *Baseline*: Neural network under normal training; *Preprocessing*: Neural network under normal training with label processed by Jiang & Nachum

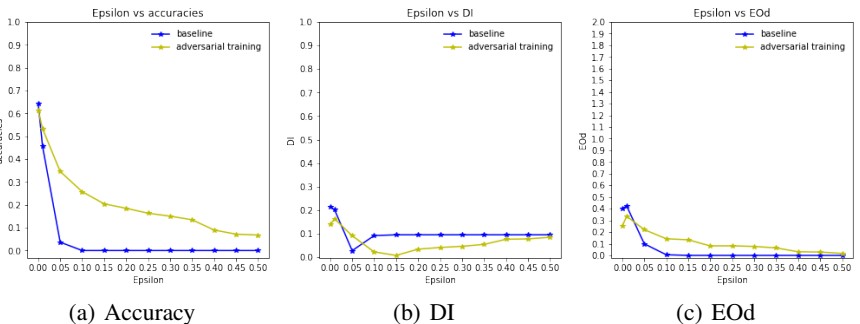

(a) Accuracy  (b) DI  (c) EOd

Figure 3: Change of accuracy, DI and EOd under accuracy attack on Adult dataset.

(2020); *In-processing*: Neural network under normal training with relaxed EOd constraint by Wang et al. (2022); *Post-processing*: Neural network under normal training with post-processing technique by Jang et al. (2022); *Adversarial training*: Neural network under adversarial training w.r.t. accuracy. We consider three different versions for our fair adversarial training method: *Adversarial training (preprocessing)*: Neural network under adversarial training w.r.t. accuracy with training labels processed by Jiang & Nachum (2020); *Adversarial training (in-processing)*: Neural network under adversarial training w.r.t. accuracy with relaxed EOd constraint by Wang et al. (2022); *Adversarial training (post-processing)*: Neural network under adversarial training w.r.t. accuracy with predicted soft labels postprocessed by Jang et al. (2022).. The three versions differ in the fairness regularization $L$ in equation 4.

Results on classifiers under fairness attack on Adult dataset are shown in Fig. 2. The fairness attack enforces biased predictions against testing samples based on the sensitive information, and under a successful attack (the DI reaches its maximum), the EOd also reaches its maximum, while the accuracy is dependent of the base rate in each sensitive group. Compared with methods under adversarial training, methods under normal training show a sharp increase in DI and EOd under fairness attacks, and improvement in fairness under normal training does not help with fairness robustness. In comparison, classifiers under adversarial training w.r.t. accuracy show improvement in fairness robustness, and classifiers under fair adversarial training show further remarkable improvement in terms of fairness robustness. We defer full results and ablation study to the appendix.

## 6.2 ROBUSTNESS AGAINST ACCURACY ATTACK

We move on to discuss the improvement of accuracy robustness under adversarial training w.r.t. fairness. We compare two different methods: *Baseline*: MLP model under normal training; *Adversarial training (DI)*: MLP model under adversarial training w.r.t. relaxed DI. We show results on classifiers under accuracy attack on Adult dataset in Fig. 3. Under a successful accuracy attack (the accuracy reaches its minimum), the EOd also becomes zero, while DI does not necessarily vanishes due to group-wise disparities in base rates. Compared with baseline, classifier under adversarial training shows remarkable improvement in accuracy robustness, which validates that accuracy robustness also benefits from adversarial training w.r.t. fairness. Results on other datasets are shown in the appendix.

## 7 CONCLUSION

Fairness attack and defense is an important yet not properly addressed problem. In this paper, we propose a unified framework for fairness attack against group fairness notions, where we show theoretically the connection of fairness attacks under different notions, and we demonstrate the connections between fairness attack and accuracy attack. We show theoretically the alignment between fairness robustness and accuracy robustness, and we propose a fair adversarial training structure, where the goal is to improve fairness robustness while maintaining fairness. Further, from experiments we validate that our method achieves better fairness robustness, and that fairness robustness and accuracy robustness align with each other. Future directions include finding alternative relaxations for fairness attack, and alternative training strategies for fair adversarial training.

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
