# OpenReview forum: "Fair Adversarial Training: on the Adversarial Attack and Defense of Fairness"
_ICLR.cc/2024/Conference — Submitted to ICLR 2024_

### Official Review · Reviewer_qsra · 2023-10-28

**Soundness:** 3 good
**Presentation:** 3 good
**Contribution:** 3 good
**Rating:** 6
**Confidence:** 4

**Summary:**

This paper studies the the problems about fairness attack and fairness defense. In particular, the authors first propose a unified framework for fairness attacks and theoretically demonstrate the alignment between fairness robustness and accuracy robustness. Then, the authors propose a novel defense framework, called fair adversarial training. and empirically validate the superiority of their proposed method.

**Strengths:**

1.	This paper is well-written, the notations and the definitions are clear.
2.	The framework for fairness attack proposed in this paper is general.
3.	The theoretical analysis is sufficient and interesting, for example, they prove the alignment between fairness robustness and accuracy robustness.

**Weaknesses:**

1.	The experimental setup is not clear.
2.	It would be better if the authors provide the code.

**Questions:**

Please refer to the Weaknesses.

---

> ### Author Response · Authors · 2023-11-18
> **Response to reviewer qsra**
>
> Thank you for the detailed comment.
>
> ### [Weakness 1 (W1): Experimental setup]
> Sorry for the confusion. Detailed experimental setup can be found in Section A of the appendix.
>
> ### [W2: Source code]
> We 'll provide the source code in final version.

---

### Official Review · Reviewer_AyY3 · 2023-10-31

**Soundness:** 3 good
**Presentation:** 2 fair
**Contribution:** 3 good
**Rating:** 6
**Confidence:** 3

**Summary:**

This paper addresses the challenge of how adversarial training performs in the context of adversarial perturbation against fairness by studying the problem of adversarial attack and adversarial robustness for fairness and accuracy. From the adversarial attack perspective, this paper proposes a unified structure for adversarial attacks against fairness, which brings together common notions of group fairness. This paper proves the equivalence of adversarial attacks against different fairness notions and derives the connection between adversarial attacks against fairness and accuracy. From the adversarial robustness perspective, this paper theoretically aligns robustness to adversarial attacks against fairness and accuracy.

**Strengths:**

1.The author addresses the problem of applying adversarial training in the context of fairness depreciation under adversarial perturbation, which is important but underexplored.

2.The paper provides a clear definition and explanation of the difference between accuracy robustness and fairness robustness.

3.This paper demonstrates that based on spatial proximity, samples can attain alignment between the fairness attack and the accuracy attack. This paper also provides a theoretic discussion about how fairness robustness and accuracy robustness can benefit from each other.

**Weaknesses:**

1. The explanation of Figure 1 is unclear. The paper claims that the Balck TPR and White TNR, with 100% accuracy, will cause destructive and social injustice against the disadvantaged group. However, it may be correct. Only discussing Black TPR and White TNR is not sufficient and not convincing. The problem for causing injustice is the model has a high Black TPR while having a low Black TNR. (or Low White TPR and High White TNR).

2. The presentation could be clearer (sentence after Equation 5, “adversarial sample ny xi”, what does ny represent?)

**Questions:**

See above.

---

> ### Author Response · Authors · 2023-11-18
> **Response to reviewer AyY3**
>
> Thank you for the detailed comment.
>
> ### [Weakness 1 (W1): Explanation of Figure 1]
> Sorry for the confusion. As shown in Fig. 1, under the fairness attack, the black TPR increases to 1, while the black TNR decreases to 0. This means all the black people are predicted as "high risk". On the contrary, the white TPR decreases to 0, while the white TNR increases to 1. This means all the white people are predicted as "low risk". Consequently, such disparity leads to crucial injustice problems.
>
> ### [W2: Presentation issue]
> "ny" should be "by" in the content. We are sorry for the typos and we 'll fix them in final version.

---

### Official Review · Reviewer_HjJ2 · 2023-11-01

**Soundness:** 2 fair
**Presentation:** 2 fair
**Contribution:** 2 fair
**Rating:** 5
**Confidence:** 3

**Summary:**

This paper proposes a unified framework for fairness attacks and defense.

**Strengths:**

This paper formally defines fairness attacks and fairness defense. The mathematical formulation is very clean.

Experiments show that fair attack can successfully attack natural training models, reaching maximum EOD and DI. While fair adversarial training successfully reduces them to 0.

**Weaknesses:**

Firstly, I recommend reformatting Corollary 1 and 2 to be Proposition 1 and 2.

The designations of Theorem 1 and 2 seem more suited to intermediate results rather than formal theorems. Specifically, regarding Theorem 1:

1) The paper doesn't explain how the fairness robustness between $x_{FN,1}$ and $x_{FN,0}$ can be employed to ascertain the fairness robustness guaranteed by accuracy robustness.

2) The upper bound presented appears convoluted. The significance and roles of each term, notably $G_t$ and $\eta_t$, are not adequately explained. Furthermore, the insights or conclusions that readers should glean from Theorem 1 and Theorem 2 ought to be more comprehensively discussed.

Similar issue appear to Theorem 2.

3) The discussion of the experiments is sparse.

4) Most importantly, the purpose of a fairness attack remains unclear to me. What specific objective does a fairness attacker want to attack the fairness of an ML model? Consequently, I question the emphasis on increasing model fairness against such attackers. Addressing fairness issues stemming from data or algorithmic biases appears to be more important.

Based on the last question, I will first give a confidence score 2. I am flexible about adjusting my score.

**Questions:**

See weakness.

---

> ### Author Response · Authors · 2023-11-18
> **Response to reviewer HjJ2**
>
> Thank you for the detailed comment. For a better clarification of our framework, we refer to the **revised version** of our paper in the following discussion. Specifically, in the revised version, Thm1&2 are about guaranteeing fairness robustness using accuracy robustness, and Thm3 is about guaranteeing accuracy robustness using fairness robustness.
>
> ### [Weakness 1 (W1), W2: Explanation of Theorem 1 and 2]
> We refer to Theorem 2 in the revised paper to show the practical implication of Theorem 1 in the revised paper regarding the fair adversarial training framework. Specifically, in the revised version, the upper-bound of fairness robustness regarding samples in the "Opposite Direction" in Theorem 1 allows us to upper-bound changes of EOd and DI under the fairness attack using static fairness, i.e., fairness issues stemming from data or algorithmic biases without the adversarial perturbtion, and accuracy robustness of the model. Consequently, Theorem 1 and 2 in the revised version provides theoretical guarantee to our fair adversarial training framework.
>
> ### [W3: Discussion of the experiments]
> Thank you for the suggestion. We 'll inlcude more analysis regarding the experimental results in final version.
>
> ### [W4: Motivation of the fairness attack]
> We agree with the reviewer that static fairness, namely fairness issues stemming from data or algorithmic biases is an important problem, and therefore our fair adversarial framework takes into account both static fairness and fairness robustness, as shown in Eq. 5. Similar to that of the accuracy attack, in terms of pracitcal applicability, it is peculiar to intentionally depreciate a model's accuracy or fairness, unless out of some egregious concern, as dicussed in Fig.1 of the main paper. Instead, the fairness attack can be considered as means of measuring of generalizability of model in terms of fairness on some unknown data under the worst-case scenario. As discussed in Section 6.2 of the main paper, under a successful accuracy attack, the accuracy of the model decreases to 0, while the DI or EOd does not show a monotonic increase as the perturbation level of the accuracy attack increases. This shows that the accuracy attack alone does not provide adequate information in terms of the generalizability regarding the fairness of the model under dynamic secnarios. Therefore, as with the accuracy attack, we propose the fairness attack to quantify the generalizability of a model in terms of fairness regarding the worst-case scenario.

---

> > ### Comment · Reviewer_HjJ2 · 2023-11-23
> > **Thanks for the thoughtful response from the authors.**
> >
> > Thanks for the thoughtful response from the authors. My main concern is about the settings. As it is stated that:
> >
> > > the fairness attack can be considered as means of measuring of generalizability of model in terms of fairness on some unknown data under the worst-case scenario.
> >
> > The setting seems very conservative to me.
> >
> > We could compare this setting with a typical adversarial scenario. Adversarial training isn't primarily done for generalizability in worst-case scenarios; rather, it's tailored for actual attack situations. If there are no real-world attacks, people will not employ adversarial training. Instead, they will use more suited techniques to enhancing generalizability in worst-case conditions.
> > Consequently, this setting still strikes me as somewhat artificial.

---

> > > ### Author Response · Authors · 2023-11-23
> > > **Follow-up to reviewer HjJ2**
> > >
> > > Thank you for the follow-up. Regarding the generability perspective of adversarial training, we may consider the objective of a typical adversarial training:
> > > $$
> > > \min \_\theta \mathbb{E}\_{(x, y) \sim \mathcal{D}}\left[\max \_{\delta \in B(x, \varepsilon)} \mathcal{L}\_{c e}(\theta, x+\delta, y)\right],
> > > $$
> > > where $\mathcal{D}$ is the training distribution and $B(x, \varepsilon):=\\{x+\delta \in \mathbb{R}^m \mid \|\delta\|_p \leq \varepsilon\\}$. Such mini-max formulation corresponds directly to the worst-case performance of the classifier under varying data distributions, i.e., the generability of the classifier under worst-case scenarios. Consequently, the adversarial attacks (namely the accuracy attacks in this case) can be thought of as means to obtain the worst-case samples [1,2]. Using the accuracy attack corresponds to the worst-case in generability in accuracy, while the fairness attack considers the worst-case in generability in fairness measures.
> > >
> > > Moreover, regarding the practical threats, fairness attacks can be carried out for malicious intentions or anarchistic reasons against the disadvantaged groups [3].
> > >
> > > [1] Bai, Tao, et al. "Recent advances in adversarial training for adversarial robustness." arXiv preprint arXiv:2102.01356 (2021).
> > >
> > > [2] Wang, Yisen, et al. "On the convergence and robustness of adversarial training." arXiv preprint arXiv:2112.08304 (2021).
> > >
> > > [3] Chhabra, Anshuman, et al. "Robust fair clustering: A novel fairness attack and defense framework." arXiv preprint arXiv:2210.01953 (2022).

---

### Official Review · Reviewer_D6Dd · 2023-11-01

**Soundness:** 1 poor
**Presentation:** 2 fair
**Contribution:** 2 fair
**Rating:** 3
**Confidence:** 4

**Summary:**

- The authors present a unified structure for adversarial training that can be applied to various notions of group fairness.

- The authors demonstrate some theoretical results that establish a relationship between robustness for fairness and robustness for accuracy.

- The authors have proposed an adversarial training scheme that considers both fairness robustness and accuracy robustness, and they have verified its performance through several experiments.

**Strengths:**

- The paper introduces a novel concern: robustness concerning fairness measures.

- The proposed algorithm and theoretical findings have been validated through experiments.

**Weaknesses:**

- The concept of an adversarial attack concerning a group fairness measure seem unconventional. While existing adversarial attacks utilize point-wise objectives, the proposed fairness attack employs a group-wise objective such as DI or EOd. When this proposed attack is applied at a single point, it can influence all subsequent attacks from other points. This leads to a notational issue in defining fairness adversarial samples: $L_{DI}$ and $L(x, a, y)$ are used interchangeably. It might be more suitable to carry out the attack using individual fairness measures.

- The presentation of fairness robustness lacks clarity in its definition. Only expressions - related to fairness robustness, indicate improved fairness robustness - are presented. The absence of a clear definition weakens the authors' assertions.

- The main assertion here is that the robustness secured by accuracy adversarial samples can guarantee fairness robustness, and vice versa. However, there appears to be a significant issue. When a fairness attack is executed with an objective like DI, its evaluation relies on cross-entropy. Then, in terms of accuracy robustness, fairness attacks essentially amount to weaker attacks on cross-entropy compared to PGD attacks with a cross-entropy objective. In light of this, the proposed claims about alignment of robustnesses become somewhat trivial, potentially diminishing the authors' contributions.

- Being robust to PGD attacks does not necessarily imply the overall adversarial robustness of the model. For a more reliable evaluation of adversarial robustness, I would encourage you to consider using AutoAttack (AA) [1]. AA comprises three white-box attacks (APGD and APGD-DLR as described in [1], and FAB in [2]), along with one black-box attack (Square Attack [3]). In the recent field of adversarial robustness research, it has become a standard practice to report robust accuracies against AA as a means of assessing gradient obfuscation [4].

- (Notational Problems) The authors are simultaneously using $D_{sub,a}^{DI}$ and $D(x_{sub,a})$. To express this more appropriately, I suggest that $D_{sub,a}^{DI}$ should be utilized as notation for group characteristics, for example: $D_{sub,a}^{DI}=\min_{sub,a}D(x)$. Furthermore, the notation $x_{sub,a}$ can be confusing, as it refers to a subgroup at times and to an individual point at other times. The lack of a formal definition for fairness robustness is connected to these issues.


[1] Croce, F. and Hein, M. Reliable evaluation of adversarial robustness with an ensemble of diverse parameter-free attacks, In ICML, 2020

[2] Croce, F. and Hein, M. Minimally distorted adversarial examples with a fast adaptive boundary attack. In ICML, 2020.

[3] Li, Y., Xu, X., Xiao, J., Li, S., and Shen, H. T. Adaptive square attack: Fooling autonomous cars with adversarial traffic signs. IEEE Internet of Things Journal, 2020.

[4] Papernot, N., McDaniel, P., Sinha, A., and Wellman, M. Towards the science of security and privacy in machine learning. 2018 IEEE European Symposium on Security and Privacy (EuroS&P), 2018

**Questions:**

- Is the term "accuracy" mentioned in the experiments referring to standard accuracy or robust accuracy? If it pertains to robust accuracy, it would be more informative to also include standard accuracy in the presentation of experimental results.

- In Table 2 in Section B of the appendix, how is the value of $D(x_{FN,male})$ calculated? Is it based on a single sample from the subgroup, or is it a group-wise value?

- Regarding the proposed algorithm, the fairness regularization is denoted by gamma. Were the experiments conducted with a fixed gamma value? I couldn't find the specific gamma value in the experimental setups. How do the results vary when gamma is changed, and does the proposed algorithm outperform other methods under different gamma settings?

---

> ### Author Response · Authors · 2023-11-18
> **Response to reviewer D6Dd**
>
> Thank you for the detailed comment. For a better clarification regarding the notation and our framework, we refer to the **revised version** of our paper in the following discussion. Specifically, in the revised version, Thm1&2 are about guaranteeing fairness robustness using accuracy robustness, and Thm3 is about guaranteeing accuracy robustness using fairness robustness.
>
> ### [Weakness 1 (W1): Formulation of the fairness attack]
> We would like to clarify that both the fairness attack and the accuracy attack employ collective adversarial objectives, rather than point-wise objectives. Regarding the poisoning attacks, the adversarial objectives take into account the statistical properties of the classifier at a collective level, such as accuracy or DI/EOd subject to the specified targets, rather than the point-wise objectives as in the evasion attacks.
>
> Moreover, similar to that of the accuracy attack, while the adversarial objective is targeted against the statistical properties of the model, i.e., DI and EOd of the model, the fairness attack on different data points are independent. This is further discussed in Eq. 2 and Eq. 3 of the main paper, which is a instance-wise reformulation of Eq. 1. While the estimation of DI and EOd relies on group information, regarding gradient-based fairness attack, the adversarial objective $L_{\text{DI}}$ only takes into account the soft predicton of $x_i$, and information from all the other points can be treated as constants, as they do not affect the update of the fairness adversarial samples by $x_i$. **Therefore, the fairness attack applied to one single instance does not influence the subsequential attacks for the others, and the adversarial attack is performed based on purely the gradient information of the trained model w.r.t. the very instance.**
>
> ### [W2: Definition of fairness robustness]
> The definition of fairness robustness is presented in the introduction of main paper: "Here we similarly define fairness robustness as a model’s ability to resist adversarial perturbation by an fairness attack and remain same predictions on clean and fairness adversarial samples.", i.e., $\mathbb{P}\_{x \sim \mathcal{D}\_{\mathcal{X}}}[\exists \|\epsilon^{\text{Fair}}\| \le \epsilon\_0$ s.t. $f(x+\epsilon^{\text{Fair}}) \neq f(x)]$ where $\epsilon^{\text{Fair}}$ is specified by the fairness attack. This reflects the consistency of the classifier to maintain the same predictions both with and without the adversarial perturbation under the fairness attack. Similarly, in the introduction, we define accuracy robustness as "a model's ability to resist adversarial perturbation by an accuracy attack and remain same predictions on clean and accuracy adversarial samples.", i.e., $\mathbb{P}\_{x \sim \mathcal{D}\_{\mathcal{X}}}[\exists \|\epsilon^{\text{Acc}}\| \le \epsilon\_0$ s.t. $f(x+\epsilon^{\text{Acc}}) \neq f(x)]$ where $\epsilon^{\text{Acc}}$ is specified by the accuracy attack. This reflects the consistency of the classifier to maintain the same predictions both with and without the adversarial perturbation under the accuracy attack.
>
> ### [W3: Alignment of robustness]
> Regarding the effect of adversarial attack, both the magnitude and the direction of the attack matter. As discussed in Section 4 of the main paper, The fairness attack and the accuracy attack acts upon different directions regarding different samples. Under the same step size and the same perturbation level up to the same iterations, the fairness attack and the accuracy attack act equivalently upon samples in the 'Same Direction' category. For samples in the 'Opposite Direction' category, the fairness attack and the accuracy attack act towards the opposite directions. We cannot assert that "weaker attacks on cross-entropy" leads to trivial alignment between fairness robustness and accuracy robustness, owing to the misalignment in the direction of the two attacks. Therefore, we show in Theorem 1 in the revised paper that the fairness robustness, measured by the change in cross-entropy loss before and after the fairness attack, regarding $x_{\text{FN},1}$ is upper-bounded by the accuracy robustness of $x_{\text{FN},0}$, i.e., the fairness robustness guarantee by accuracy robustness. Moreover, our discussion in Theorem 3 in the revised paper also points to the non-trivialness of the problem: if the fairness robustness amounts to weaker robustness than the accuracy robustness, we cannot have the accuracy robustness guarantee by the fairness robustness.

---

> ### Author Response · Authors · 2023-11-18
>
> ### [W4: Results of AutoAttack]
> Thank you for the suggestion. We show results using AutoAttack [1] to further validate the effectiveness of our method:
>
> Method  | Accuracy | EOd | DI
> -------------------|------------------|------------------|------------------
> Baseline| 31.75 $\pm$ 0.57\% | 200\% | 100\%
> Preprocessing| 31.75 $\pm$ 0.57\% | 200\% | 100\%
> In-processing| 33.67 $\pm$ 0.74% | 194.73 $\pm$ 2.26\% | 91.24 $\pm$ 1.33\%
> Post-processing| 31.75 $\pm$ 0.57\% | 200\% | 100\%
> Adversarial training | 72.67 $\pm$ 0.88\% | 62.47 $\pm$ 2.63\% | 36.58 $\pm$ 2.19\%
> Ours: adversarial training (preprocessing) | 75.67 $\pm$ 0.76\% | 24.47 $\pm$ 1.47\% | 15.49 $\pm$ 1.35\%
> Ours: adversarial training (in-processing)| 74.35 $\pm$ 0.55\% | 27.35 $\pm$ 2.21\% | 17.12 $\pm$ 1.14\%
> Ours: adversarial training (post-processing)| 74.67 $\pm$ 0.45\% | 28.17 $\pm$ 1.77\% | 17.43 $\pm$ 1.26\%
>
> The table above shows results of AutoAttack (the objective of white-box attacks are replaced by $L_{\text{DI}}$ the objective of fairness attack) on Adult dataset under perturbation level $\epsilon=0.2$. Compared with the baselines, our methods show remarkable improvement in terms of fairness robustness. We 'll include full results on other datasets in final version.
>
> ### [W5: Notational issues]
> Thank you for the detailed suggestion. Please refer to the revised version for a more comprehensive version of notations.
>
> ### [Question 1 (Q1): Accuracy in the experiments]
> The accuracy in the experiments refers to both standard accuracy and robust accuracy. As shown in Fig. 2 and 3, when the perturbation level $\epsilon$ equals zero, it corresponds to the standard accuracy results. Under non-zero perturbation levels, it gives the robust accuracy results.
>
> ### [Q2: Calculation of $D(x_{\text{FN},female})$]
> The values in Table 2 are calculated based on the group-wise average. For example, $D(x_{\text{FN},female}) = \frac{1}{|\{\text{FN},female\}|}\sum_{x_i \in \{\text{FN},female\}} D(x_i)$.
>
> ### [Q3: Setting of $\gamma$]
> The $L$ term in Eq. 5 corresponds to the fairness measures, i.e., DI or EOd in our discussion. It is hard to directly set $\gamma$ as some constant during training, as the fairness measures are intractable and non-differentiable. Instead, during training, $\gamma$ is integrated as a Lagrangian multiplier as a fairness regularizer where the multiplier value varies by the methods, and the hyperparameters of different methods (preprocessing, in-processing and post-processing) are tuned as suggested by the authors in the original paper to find the best performance in terms of fairness.
>
> [1] Croce, F. and Hein, M. Reliable evaluation of adversarial robustness with an ensemble of diverse parameter-free attacks, In ICML, 2020

---

> > ### Comment · Reviewer_D6Dd · 2023-11-23
> >
> > I appreciate the points raised in the rebuttal. The authors' research findings are inherently dependent on the definitions of both accuracy robustness and fairness robustness. While the authors presented definitions for both in the rebuttal, there are no theoretical results in this paper that directly utilize this definition. To make this paper more meaningful, the main theorem should establish inequalities that connect the well-defined accuracy robustness with the well-defined fairness robustness. Moreover, the authors' other definitions are also unclear (e.g. using mixed notations for a sample and a subgroup) as I previously mentioned, making it difficult to even evaluate. Therefore, currently, the authors' arguments are weak and disorganized. Despite my appreciation for the authors' efforts, I will maintain my current rating.

---

> > > ### Author Response · Authors · 2023-11-23
> > > **Follow-up to reviewer D6Dd**
> > >
> > > Our discussions in Theorem 1 and Theorem 3 in the **revised paper** provides instance-level utilization of the definitions regarding fairness robustness and accuracy robustness in the rebuttal. Owing to the disparities in the directions of fairness attack and accuracy attack, it is more meaningful to consider the instance-wise relationships betweeen fairness robustness and accuracy robustness on samples in the "Opposite Direction" category of Table 1, rather than on the whole dataset. Moreover, our discussion in Theorem 2 in the **revised paper** establishes inequality that connects fairness robustness and accuracy robustness. We 'll further revise the notations to better distinguish between samples and subgroups in final paper.

---

> > > > ### Comment · Reviewer_D6Dd · 2023-12-04
> > > >
> > > > As previously noted, using a single notation with multiple meanings can be problematic as it can change the authors' arguments themselves. In response to Question 2 in our initial review, the authors stated that $D$ represents group-wise average. However, upon reading the proof process in the **revised paper**, it appears that the $D$ used in Theorem 1 of the **revised paper** is a sample-wise value. Furthermore, the definition of fairness robustness presented by the authors in their rebuttal is described from the perspective of the entire dataset. In other words, it seems that the authors have used a single notation with different meanings as needed. Without clarification, the meaning of Theorem 1 and 3 presented by the authors in the revised paper may differ from the bound of fairness robustness they defined.
> > > >
> > > > As mentioned earlier, the fairness robustness defined by the authors pertains to the perspective of the entire dataset, and thus, an appropriate upper bound in line with this perspective is necessary. This upper bound should also be described from the perspective of the entire dataset, with discussion about its suitability. I hope you understand that this review is not solely based on the issue of notation but also concerns the potential change in the meaning of the arguments due to this issue.
> > > >
> > > > Because of this notational problem, it is challenging to grasp the concept, motivation, and theoretical results of this paper when reading it. I genuinely appreciate the authors' efforts to support their claims with various experiments. I find this research to be highly interesting and believe it will yield meaningful results when supported by precise foundations. However, due to the reasons mentioned above, I hope that this research will further develop beyond its current state, and I will maintain my score accordingly.

---

### Meta-Review · Area_Chair_HQ97 · 2023-12-11

**Metareview:**

The paper introduces a framework for understanding and countering adversarial attacks that aim to impact group fairness guarantees in machine learning models. The results indicate a close connection between adversarial fairness and accuracy attacks, demonstrating that robustness to one can enhance robustness to the other. The paper also introduces an approach to "fair" adversarial training that integrates fair classification with adversarial training, improving resilience against adversarial perturbations targeting both fairness and accuracy. The theoretical results point towards a connection between various notions of group fairness and elucidate how adversarial attacks on fairness and accuracy are interrelated.

Strengths of the paper include the compelling problem setting and the significant numerical results.

However, as noted by the reviewers, the paper has several limitations. Reviewer D6Dd raised multiple concerns. The reviewer highlighted the inconsistent use of notation in the paper, potentially altering the meaning of key arguments. These ambiguities complicate the understanding of the main theorems as well as the bounds of fairness robustness. The reviewer also emphasized that the critique extends beyond mere notational issues and fundamentally impacts the paper, remaining cautious after the discussion period.

Reviewer HjJ2 also stood by concerns regarding the notation used in the paper and the core motivation behind the introduced fairness attacks. Note that the paper's presentation (notation, theorems, problem setting) was a concern shared across reviewers.

The paper addresses an interesting problem, but the motivation, notation, and overall presentation have to be significantly revised prior to acceptance. I also note that the authors miss several state-of-the-art fairness interventions in their benchmarks, such as

* Lowy, A., Pavan, R., Baharlouei, S., Razaviyayn, M., & Beirami, A. (2021). Fermi: Fair empirical risk minimization via exponential Rényi mutual information.
* Wei, D., Ramamurthy, K. N., & Calmon, F. P. (2021). Optimized score transformation for consistent fair classification. The Journal of Machine Learning Research, 22(1), 11692-11769.
* Agarwal, Alekh, Alina Beygelzimer, Miroslav Dudík, John Langford, and Hanna Wallach. "A reductions approach to fair classification." In International conference on machine learning, pp. 60-69. PMLR, 2018.

Given the missing literature and opaque notation and problem motivation, this paper will benefit from further revisions.

**Justification For Why Not Higher Score:**

The paper could benefit from significant revision of the notation, theoretical results, related literature, and motivation (as suggested in the reviews).

**Justification For Why Not Lower Score:**

N/A

---

### Decision · Program_Chairs · 2024-01-16

Reject